# Energy Transitions Towards Low Carbon Resilience: Evaluation of Disaster-Triggered Local and Regional Cases

**Yekang Ko** [1]**, Brendan F. D. Barrett** [2,*]**, Andrea E. Copping** [3]**, Ayyoob Sharifi** [4,5]**, Masaru Yarime** [6,7,8] **and Xin Wang** [9]

1   School of Architecture and Environment, University of Oregon, Eugene, OR 97403, USA; yekangko@uoregon.edu
2   Center for the Study of Co*Design, Osaka University, Toyonaka 560-0043, Japan
3   Pacific Northwest National Laboratory and University of Washington, Seattle, WA 98019, USA; Andrea.Copping@pnnl.gov
4   Graduate School of International Development and Cooperation, Hiroshima University, Higashi-Hiroshima 739-8530, Japan; sharifi@hiroshima-u.ac.jp
5   Network for Education and Research on Peace and Sustainability (NERPS), Hiroshima University, Higashi-Hiroshima 739-8530, Japan
6   Division of Public Policy, Hong Kong University of Science and Technology, Hong Kong SAR, China; yarime@ust.hk
7   Department of Science, Technology, Engineering and Public Policy, University College London, London WC1E 6JA, UK
8   Graduate School of Public Policy, The University of Tokyo, Tokyo 113-0033, Japan
9   College of Architecture and Urban Planning, Tongji University, Shanghai 200092, China; xin_wang@tongji.edu.cn
*   Correspondence: brendan.barrett@cscd.osaka-u.ac.jp; Tel.: +81-6-6850-8321

**Abstract:** Following numerous global scientific studies and major international agreements, the decarbonization of energy systems is an apparent and pressing concern. The consequence of continued emission growth tied to rising global average temperatures is difficult to predict, but against a background of other natural and human-induced disasters, may create a situation, from a positive perspective, where each disaster event triggers "build back better" responses designed to speed the transition toward low carbon, resilience-oriented energy systems. This article examines the potential for disaster-triggered responses in communities, at various local and regional levels, in four industrial economies in the Asia Pacific region: Japan, China, Australia, and the USA. Seven case studies were evaluated against a set of criteria that exemplify the key aspects of resilient energy systems. The research results suggest that a new space of innovation does emerge in post-disaster situations at a range of local and regional scales. The greatest potential benefit and opportunity for significant gains, however, appears to manifest at the small community level, and the ultimate challenge relates to how to mainstream local innovations into state and national level transformation on energy systems so as to enhance resilience and promote rapid decarbonization.

**Keywords:** climate change; resilience; energy; disasters; transitions; community; local; sub-national; decarbonization; innovations; pathways; triggers

## 1. Introduction

The aim of this article is to examine how local and regional pathways towards decarbonization, and the attainment of resilience-oriented energy systems, may be triggered in the context of natural

disasters. This is elaborated through case studies from the industrial economies of Australia, Japan, China, and the USA evaluated against criteria used to measure the shift from conventional to non-conventional, resilient energy systems [1]. There is a growing body of work portraying subnational progress as fundamental to the rapid upscaling of decarbonization scenarios, with an energy transition at the core (renewable and energy efficiency), and with measures to mitigate and adapt to abrupt disasters as well as those that are slow-moving such as climate change. Recent publications suggest that local resilience to climate change and extreme weather events will be central to the ongoing transformation, especially in the Asia Pacific region. According to the United Nations, Asia and the Pacific remains the world's most disaster-prone region. From 2014 to 2017, the region experienced 217 storms and cyclones, and 236 cases of severe flooding [2]. These disasters can act as triggers (drivers, stressors and disruptors) that accelerate change in circumstances when building back better means pursuing smart, low carbon development patterns based on resilient and renewable energy systems. However, they also put in jeopardy measures designed to achieve a rapid energy transition.

The scientific imperative for rapid decarbonization of the global economy is clear. The 5th Assessment Report of the Intergovernmental Panel on Climate Change (IPCC) specified that in order for average global temperature rise to remain under 2°C, greenhouse gas emissions must peak by 2020, and then decline rapidly [3]. The subsequent 2018 IPCC Special Report reiterated this concern and highlighted pathways that involve overshooting and returning below 1.5°C during the 21st century, specifically indicating that net global $CO_2$ emissions need to fall by about 45% from 2010 levels by 2030, and reach "net zero" by around 2050 in order to avoid the 1.5°C overshoot (IPCC, 2018). There is a real need to increase our collective level of ambition, and a review of 11 global decarbonization scenarios developed between 2005 and 2013, with a focus on the required energy systems transformation, revealed that they superficially address the technical, economic, infrastructural and societal factors that may constrain a rapid energy transition [4].

At the national level, decarbonization pathways have been prepared for sixteen countries indicating that the most ambitious emission reduction scenarios would reduce aggregate emissions in 2050 by 56% compared to 2010 levels [5]. This suggests that the 1.5°C target overshoot is inevitable without a significant acceleration in the implementation of decarbonization measures. At the local and regional levels, the implications of the IPCC special report have been discussed in the context of how urban areas across the globe can pursue 1.5°C-consistent pathways for climate change mitigation and adaptation [6]. The rapid adoption of renewable energy is considered critical to the attainment of deep decarbonization with one study presenting road-maps to reach 100% renewable energy in 50 states in the USA by 2050 [7]. The economic and social benefits of investment in renewable technologies for pursuing rapid decarbonization of urban regions have been examined identifying 16 pathways related to energy efficiency in buildings, transportation and waste management [8]. Similarly, plans and effects of decarbonizing large stretches of non-urban landscapes [9], including coastal areas [10,11] have been examined.

While the significance of renewable technologies and policies for decarbonization at the local level is widely recognized, a more systematic evaluation of case studies is imperative to gain a better understanding of the scales, patterns, and processes of adopting renewable energy technologies at the local and regional scales. In addition, in the face of increasing uncertainties and frequent disasters, it is essential to understand how disasters trigger energy transitions in different geopolitical contexts and to ensure that renewable energy technologies also provide resilience co-benefits. This is an issue that, to the best of our knowledge, has not yet been addressed in the literature. Existing research provides a useful conceptual basis for assessing the resilience of renewable energy systems at the local scale [12,13]. However, these conceptual underpinnings need to be substantiated with indicative case studies that provide an enhanced evidence base to support the implementation of low carbon policies and investments. Some work has already been done on this, including case study analysis showing how extreme events may trigger transition toward resilience [14]. This article contributes to an important area of research since more systematic assessments of case studies are needed to provide

a better understanding of the opportunities that extreme events may provide for enhancing energy resilience. The case study approach lies at the heart of the methodology adopted for this research. These have been selected based on contextual familiarity and access to data. They are assessed through the application of a framework around resilience and socio-technical transition conceptualizations that have been applied to develop a set of criteria for the evaluation of the extent to which the case studies (in a post-disaster state) have moved closer to the characteristics of resilience-oriented energy systems. This article begins by presenting the conceptual framework before elaborating on key differences between conventional and resilience-oriented energy systems. Seven case studies are then summarized and evaluated against a set of 15 criteria. This is followed by a discussion of the research findings and a reflection on some of the key limitations as well as the scope for future research.

## 2. Conceptual Underpinning and Analytical Framework

The framework for analysis underpinning this article links two theoretical areas: (1) resilience conceptualizations and (2) socio-technical transitions/complex systems change.

### 2.1. Resilience Conceptualizations

Arguably one of the most commonly-used concepts in scientific and policy debates on climate change, resilience has its roots in the engineering, psychology, and ecology disciplines, and has gained traction in local and regional studies over the past two decades. The concept is highly emphasized in international frameworks such as the New Urban Agenda [15], the UN Sustainable Development Goal No. 11 [16], and the Sendai Framework for Disaster Risk Reduction [17]. The focus of this article is on three dominant conceptualizations in terms of engineering, ecological, and socio-ecological resilience.

*Engineering resilience (EngR)* emphasizes the development of fail-safe systems that can experience shocks without their functionality being significantly impacted. In instances where functionality is minimally impacted, engineering resilience requires rapid recovery to an equilibrium (pre-disturbance) state. Accordingly, it can be described as a rigid conceptualization of resilience [18].

*Ecological resilience (EcoR)* places emphasis on designing systems in a way that they can undergo disruption and still maintain essential functionalities and underlying relationships that govern the system [19]. The ability to absorb shocks is essential for achieving ecological resilience and facilitates a smoother recovery that may not necessarily be to the pre-disturbance equilibrium conditions. In other words, the system may return to a new equilibrium state(s) on the condition that its structure and functions remain unchanged [18,20,21].

*Socio-ecological resilience (SocR)* (also referred to as adaptive resilience) emerged in recognition of the increasing uncertainties in future conditions due to a combination of forces such as climate change and rapid population growth. This conceptualization emphasizes notions such as 'living with risk', and 'bouncing forward', and does not require returning to equilibrium conditions. Qualities such as self-organization, bottom-up and decentralized governance, and learning from disaster are considered central to the socio-ecological conceptualization of resilience [13,22,23].

In Table 1, these conceptualizations are associated with the different characteristics of conventional and resilience-oriented energy systems.

### 2.2. Socio-Technical Transitions

Conceptualizations of how socio-technical transitions *(SocT)* occur can shed light on the way that resilient-oriented energy systems emerge via complex, non-linear, contested processes characterized by difficult negotiations and extensive coalition formation between diverse interests [24]. The central proposition is that energy transitions come about through interactions between processes at three levels as presented in Figure 1 [24,25]. First, niche innovations build up internal momentum, through learning processes, price/performance improvements and support from powerful groups. Second, changes at the landscape level create pressure on the existing regime. Third, the destabilization of the regime creates windows of opportunity for niche innovation [26]. This destabilization is not normally

associated with disaster-related situations but occurs through a process where niche management works to overcome vested interest in conventional technologies and systems—for instance, promoting solar over coal [27]. This involves horizon thinking whereby energy transitions are envisaged along a time-line that connects technical innovation to overall development patterns and ultimately, behavioral change [28].

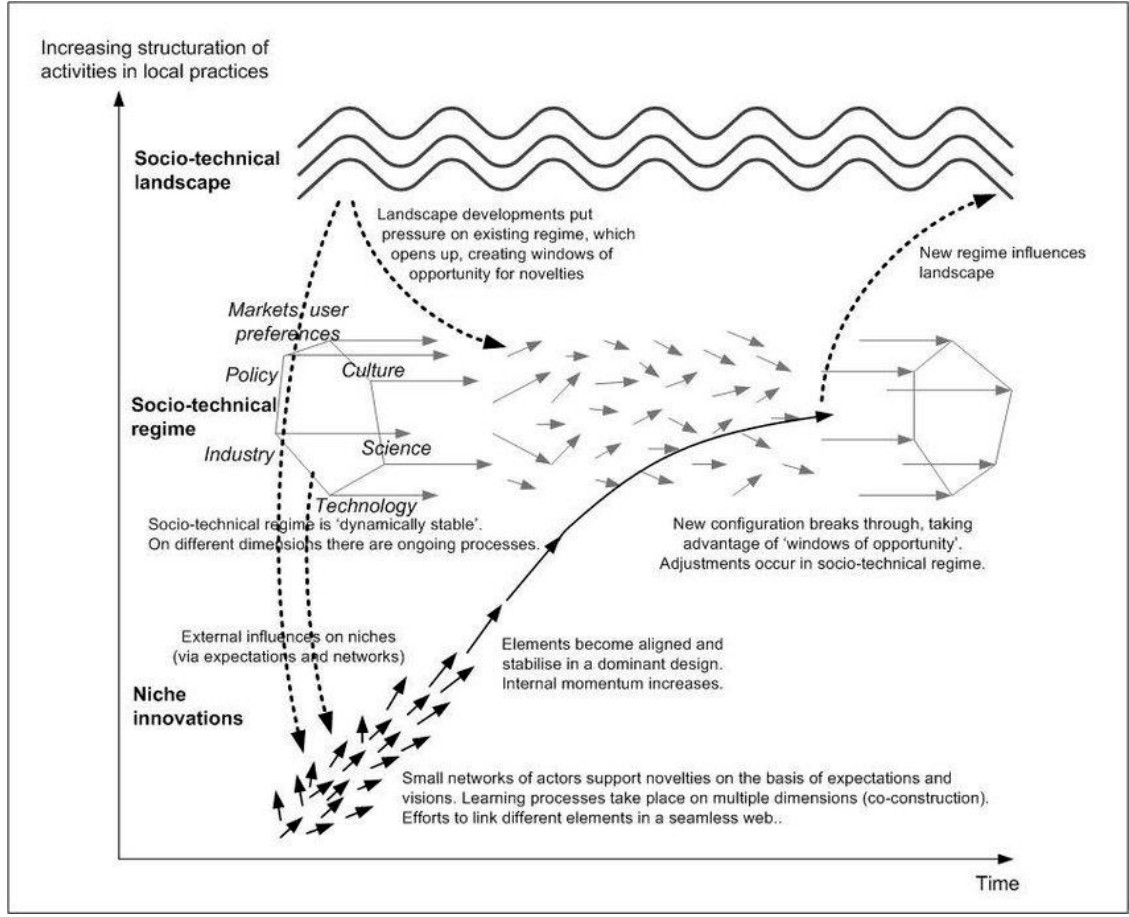

**Figure 1.** Multi-Level Perspectives (MLP) on Socio-Technical Transitions [24,25].

Energy transitions need to be understood from a systemic-evolutionary viewpoint driven by innovation policies with various pathways explored simultaneously so as to avoid lock-in [29]. The multi-level perspective (MLP) is an important factor in helping to interpret transitions where niche innovations gradually evolve into the new socio-technical regimes characterized by micro to macro changes (i.e., individual solar installations on homes, become community networks, become part of virtual power stations, and eventually become a solar city) shaped by policies, technologies and market preferences [26,30]. MLP can also be applied to the assessment of specific policies, for instance, those associated with the transition to a low carbon economy [31]. As shown in Table 1, the notion of socio-technical transitions (*SocT*) is a defining factor characterizing resilience-oriented energy systems.

### 2.3. Conventional and Resilience-Oriented Energy Systems

When applying the above conceptualizations of resilience and socio-technical transitions, it is possible to differentiate between conventional and resilience-oriented energy systems (see Table 1). Eleven characteristics are presented in Table 1 and these are cross-referenced with the three dominant conceptualizations in terms of engineering (*EngR*), ecological (*EcoR*) and socio-ecological resilience (*SocR*), as well as with the notion of socio-technical transitions (*SocT*). Resilience conceptualizations make clear why such energy systems are desirable, and socio-technical transitions theory explains how they

can be achieved, often in the face of opposition from vested interests supporting conventional, incumbent energy systems. There is an implicit assumption in Table 1 that conventional energy systems are in line with engineering resilience thinking. The other assumption here is that resilience-oriented energy systems are based on low-carbon and renewable sources with minimal detrimental environmental impacts [32,33]. Unlike conventional energy systems, resilience-oriented energy systems are diverse in size and type making them adaptable to changing circumstances and helping to avoid lock-in to undesirable pathways [34]. This suggests that their characteristics align more closely with ecological and socio-ecological resilience conceptualizations and socio-technical transitions theory.

**Table 1.** Comparison of conventional and resilience-oriented energy systems.

| Aspect/feature | Conventional Energy Systems | Resilience-Oriented Energy Systems | References |
|---|---|---|---|
| System structure | Corporate-based systems characterized by monopolistic and concentrated ownership. Development and maintenance are dependent on large-scale capital investment. Mainly the *EngR* conceptualization. | Cooperative-based networks of small-scale entities characterized by bottom-up stakeholder engagement and market competition. The more distributed power structure and higher dependence on diverse patterns of financing. *EcoR, SocE* and *SocT.* | [18,33,35] |
| Users/ownership | Users (citizens) are passive. *EngR* and *EcoR.* | A large share of technologies owned by the community, giving people more power over their own energy. *SocE* and *SocT.* | [18,33,35] |
| Energy source/Environmental impact | Mainly fossil-nuclear and hydroelectric energy systems with adverse environmental costs and effects. *EngR.* | Mainly based on low-carbon, environmentally-friendly and clean renewable sources with minimal environmental impacts. *EcoR, SocE* and *SocT.* | [32,33] |
| Adaptability | Inflexible/rigid energy systems that result in lock-in. *EngR.* | Flexible energy systems adaptable to changing circumstances. *EcoR, SocE* and *SocT.* | [18,32] |
| Innovation | Conservative systems dominated by vested interests with limited willingness to change. *EngR* and *EcoR.* | Constantly in transition and encouraging innovation supported by new coalitions. *SocE* and *SocT.* | [18] |
| Interconnectivity/interoperability | Silo-based and isolated from other systems. *EngR.* | Importance of feedback loops between different sub-systems and characterized by interconnectivity and interoperability between different components that facilitate real-time supply-demand management. *SocE.* | [18,36] |
| Modularity | Centralized and large-scale generation and transmission networks and technologies. *EngR.* | An interconnected network of small-, medium-, large-scale systems characterized by a more modular (autonomous), decentralized and distributed generation and transmission infrastructure. *EcoR, SocE* and *SocT.* | [18,35] |
| Redundancy | Limited redundancy to maximize operational and economic efficiency. *EngR.* | Redundant facilities and redundant capacity (e.g., storage capacity) to deal with uncertainty and buffer against generation variability. *EcoR, SocE* and *SocT.* | [33,36] |
| Diversity | Limited diversity. *EngR.* | Diverse socio-economic and technological structures based on a variety of energy sources. *EcoR, SocE* and *SocT.* | [18] |
| Level of establishment | Well-established systems. Private investors may prefer these systems because of concerns over return on investments. *EngR.* | Lack of commercialization. Unfamiliarity and limited trust from private investors. Need for public and governmental support (specifically, subsidies are needed for start-ups). This can make non-conventional energy systems less affordable and also vulnerable to financial crises. Therefore, alternative systems are expanding at a slow pace. *EcoR, SocE* and *SocT.* | [32] |
| Efficiency | Emphasis on economic efficiency. *EngR.* | Balancing economic efficiency with diversity and redundancy in order to enhance adaptive capacity and minimize vulnerability to surprise shocks and extreme events. However, more improvements in terms of economic efficiency are expected to be achieved as more innovative technologies will emerge in the near future. *EcoR, SocE* and *SocT.* | [37] |

Avoiding lock-in enables constant transition and facilitates the uptake of innovative and disruptive technologies that may enable better responses to changing conditions. Achieving the latter also depends on strengthening interconnectivity between different system sub-components and at multiple system levels [37]. An important difference between conventional and resilience-oriented energy systems relates to governance and operation structures. Unlike conventional systems, resilience-oriented ones follow diverse, distributed, and decentralized governance and operation structures, wherein engagement and cooperation of different stakeholders are encouraged and where new coalitions emerge [24]. For example, resilience-oriented energy systems may take advantage of digital technologies to encourage citizens to participate in the generation, management, and blockchain-enabled distributed energy trading (instead of just being passive end-users) [34,38]. This facilitates a competitive environment and provides opportunities for stakeholder engagement [33,35].

One positive result of the resilience-oriented approach is that a large share of the energy infrastructure is owned by the community. In fact, this sense of ownership is widely believed to contribute to resilience. Modularity is another noteworthy characteristic of a resilience-oriented energy system [39]. Unlike conventional ones that are characterized by large-scale and centralized infrastructure, resilient systems provide an interconnected network of small-, medium-, and large-scale infrastructures that feature a certain degree of autonomy and are capable of protecting the system from domino-effects. Such isolation also enables them to provide support to affected sub-components if needed. Despite these benefits, energy resilience may involve trade-offs. For instance, redundancy may be achieved at the cost of efficiency [36]. It is essential to develop strategies to minimize such tradeoffs.

## 3. Case Studies

Case studies from Australia, China, Japan, and the USA were selected by the researchers based on familiarity with the local context and data availability. The aim was to select case studies that demonstrate how local and sub-national resilience-oriented energy systems are triggered, either as a response to mega-scale natural disasters or as a preparedness measure to mitigate against future disaster events. A range of case studies were reviewed by participants in the Urban Renewable Energy Workshop at the 2018 APRU Sustainable Cities and Landscape Conference hosted at the University of Hong Kong. The characteristics of the selected case studies are shown in Table 2. The cases represent various types of renewable energy technologies from different geopolitical contexts in the Asia Pacific region. In addition, the cases cover different renewable energy types triggered by various environmental disasters across spatial and temporal scales, as well as policy and measures. They also represent a wide range of geographic scales at local and regional levels, from a small village with less than 100 people, to an entire island over 3 million people. The case studies span the space of temporal scales for disasters, from the slow-moving changes in climate to large seasonal storms to catastrophic seismic events that provide no warning. It is acknowledged that these case studies are limited to developed industrial economies that have the technical resources, infrastructure and funds available to facilitate relatively rapid recovery. Future research should focus on cases in developing economies in the Asia-Pacific region (e.g., Myanmar, Sri Lanka, Nepal, Vietnam, Bangladesh, Thailand, etc.) that may face greater challenges in disaster-related responses due to resource constraints. Each case study is described briefly below. Subsequently, in the next section, we evaluate the cases against the framework introduced in Table 1 to examine their alignment with the resilience-oriented principles.

**Table 2.** Selected case and their key attributes.

| | Geographic Typology | Geographic Scale | Disaster/Stressor Type (Trigger) | Renewable Energy Type | Increased Renewable Energy | Status (Temporal) | Technologies | Policy and Measures |
|---|---|---|---|---|---|---|---|---|
| South Australia, Australia | Inland and coast | Region (1.7 million population) | Major storm knocked out 23 transmission pylons in 2016 | Solar and wind | 75% RE by 2025 (currently around 42%) | On-going | Large scale battery facilities, new solar plants, virtual power plant. Planned grid interconnector. | Government investment, private sector. |
| Higashi-Matsushima, Japan | Coastal, rural | Community (43,000 population) | Tsunami after earthquake in 2011 | Solar | 29 times more solar photovoltaics than pre-disaster | Completed | Smart disaster prevention ecotown (mega-solar, biodiesel, large scale battery storage) | Government subsidies, private sector, not-for-profit. |
| Puerto Rico, USA | Island; urban and rural | Region (3.3 million population) | Hurricane Maria in 2017 | Solar and wind | Prior Maria, 20% RE goal by 2035. Now the recommendation is increasing RE and reaching 50% (4000 MW) by 2035. | On-going. To be completed by 2027. | Microgrids for critical infra and remote communities, smart grids, battery storage, onsite backup generation, combined heat and power systems | Government subsidies, private sector; US $17.6 billon. |
| Igiugig village, USA | Coastal, islands, rural | Small villages (50–200 people) | Isolation, inability to receive shipments by sea due to winter storms. Fuel for electricity (diesel) is expensive, risk of poor air and water quality. Goes against native/local ethos. | River turbines | 100 kW to 5 MW | Demonstration completed; installation under construction. | Hydrokinetic river turbine | Government subsidies, indigenous people, private sector. |
| Oregon coast, USA | Coastal, islands, rural | Region (30,000–100,000 population) | Subduction zone earthquake, tsunami risk, severe storms | Wave, wind, solar | 5–10 MW, 50% RE by 2040 | Planning | Wave energy devices | Government investment, private sector |
| Qinghai Province, China | Rural, desert Gobi | Region (5.98 million people), PV station 24.33km$^2$ | Environment disaster, trade frictions, climate negotiation | Solar PV-Hydro | 850MW PV | Completed in 2015 | Solar PV-Hydro Hybrid, 24.7 billion m3 storage capacity | Government investment, US $1.2 billon for PV |
| Urawa-Misono, Saitama City, Japan | Inland, urban | City district (7476 population) | Disruptions due to natural disasters such as earthquakes and typhoons | Solar PV with batteries | 100 kW PV | Demonstration on-going | Blockchain-based distributed energy systems with peer-to-peer exchanges | Government financial support, private investment |

### 3.1. Description of the Selected Cases

### 3.1.1. South Australia

On 28 September 2016, a once in 50-years storm involving two tornadoes resulted in damage to three electricity transmission lines in South Australia. This triggered a chain of events, with nine wind farms suffering significant power reductions, resulting in the failure of the grid interconnector with the adjacent state [40]. According to the Australian Energy Market Operator (AEMO), the South Australian network became "islanded" and unable to import electricity to make up for power reductions; and electricity supply was lost over a period of hours and days [41]. Two additional blackouts were experienced as a result of environmental factors. In December 2016, severe storm damage affected 300 transmission lines, with 155,000 properties losing power. Further, in February 2017, another blackout occurred for 90,000 properties as a result of a major heatwave. The September 2016 incident triggered political debate around the resilience of electricity grids that include a large proportion of renewable energy [42]. South Australia has the largest amount of installed wind and solar capacity (1831 MW) of all states in Australia, with renewables accounting for 43.4% of electricity production in 2017 [43]. However, the storm, and not the proportion of renewables, was to blame for the blackout [44]. Subsequently, in February 2018, the South Australian Government, controlled by the Labour Party, indicated that it would raise its renewable energy target to 75% by 2025 (although opposed by the subsequent Liberal-controlled Government), and a series of mitigation measures were proposed to increase the resilience of the electricity network [45]. These included the development of a 100 MW lithium-ion battery (the world's largest) facility by Tesla, located at Hornsdale (a major wind farm site) that commenced operation in December 2017. Another major commitment involves the development of a virtual power plant; basically, a network of home solar photovoltaic installations and battery systems working together to generate, store and feed energy into the grid. The final major enhancement to the South Australia electricity network involves the construction of a new interconnector with the grid in New South Wales, with a hefty price tag of close to AUD$1.5 billion, with a potential completion date by 2024 [46]. Projections indicate that South Australia could source the equivalent of 100% of its electricity from renewables by as early as 2025, based on AEMO's most recent Integrated Systems Plan as shown in Figure 2.

### 3.1.2. Higashi-Matsushima City

The Great East Japan earthquake and tsunami of 11 March 2011 had a devastating impact on this community with 65% of the town inundated. A total of 1134 people lost their lives and the population declined from 43,142 before the accident to 39,518 in 2016 [48]. Rebuilding efforts took shape quickly, with the publication of formal guidelines for reconstruction in December 2011 and a national government designation of Higashi-Matsushima as Environmental Future City (one of eleven designated across Japan) [49]. The aim was to "build back better" by localizing electricity supply through new renewable energy projects, including large-scale storage battery development. The municipality published a local energy vision in February 2013 with the aim of increasing energy efficiencies, enhancing renewable energy potential and reducing $CO_2$ emissions [50]. The goal was to increase the local energy supply to 44 MW by 2023 through a range of projects including mega-solar, PV installations on housing, wind turbines, and biomass. In 2009, it was estimated that local per capita $CO_2$ emissions were in the region of 4.1 tons, compared with an average of 9.5 tons per capita for Japan. The aim is to reduce per capita $CO_2$ emissions in the town to 3 tons by 2022. In the period 2014–2016, the municipal authority began promoting the concept of the Smart Disaster Eco-Town, setting up a microgrid providing electricity to 85 local houses, four hospitals and a public facility [51,52]. The scheme was completed in 2016 at a total cost of around US$4 million (see Figure 3). The main power supply is from a photovoltaic facility (400 kW), supported by a biodiesel generator (500 kVA) with a 480 kWh battery storage facility [53]. The entire energy system is managed by a local, not-for-profit and funds from this venture are reinvested back into the community. It is estimated that before the

disaster, the city was generating annually around 800 kw of electricity locally. This increased to 23,328 kw in 2018, which represented a 29-fold increase in local electricity generation. It is also asserted that locally supplying electricity is more efficient than the past centralized system, and that this has brought an additional benefit in terms of $CO_2$ emission reductions in the order of 307 tons per annum. The municipality aims to ensure that 25% of local energy requirements for residents will be from renewables as rapidly as possible [54].

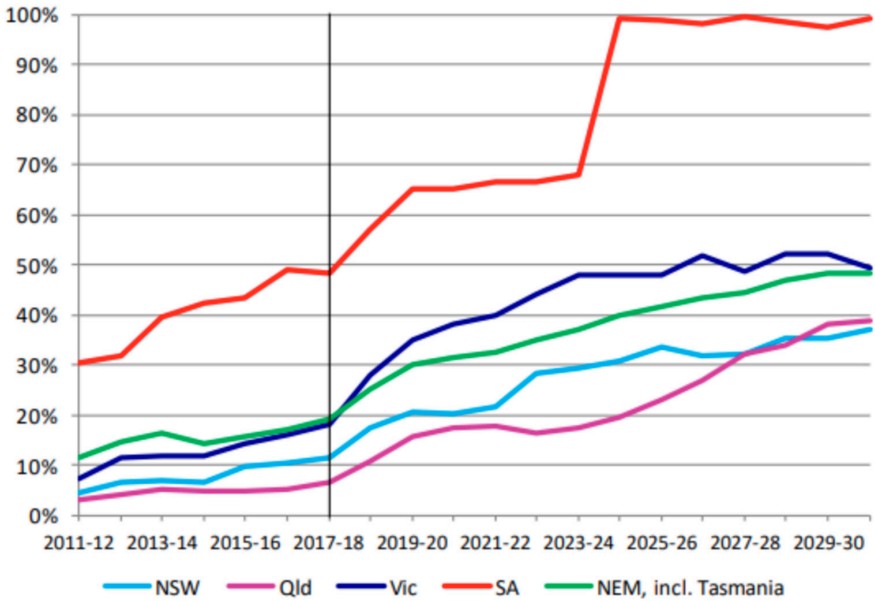

**Figure 2.** South Australia predicted to reach 100 percent renewables by 2025 [47]. NSW—New South Wales, Old—Queensland, Vic—Victoria, SA—South Australia, NEM—National Electricity Market.

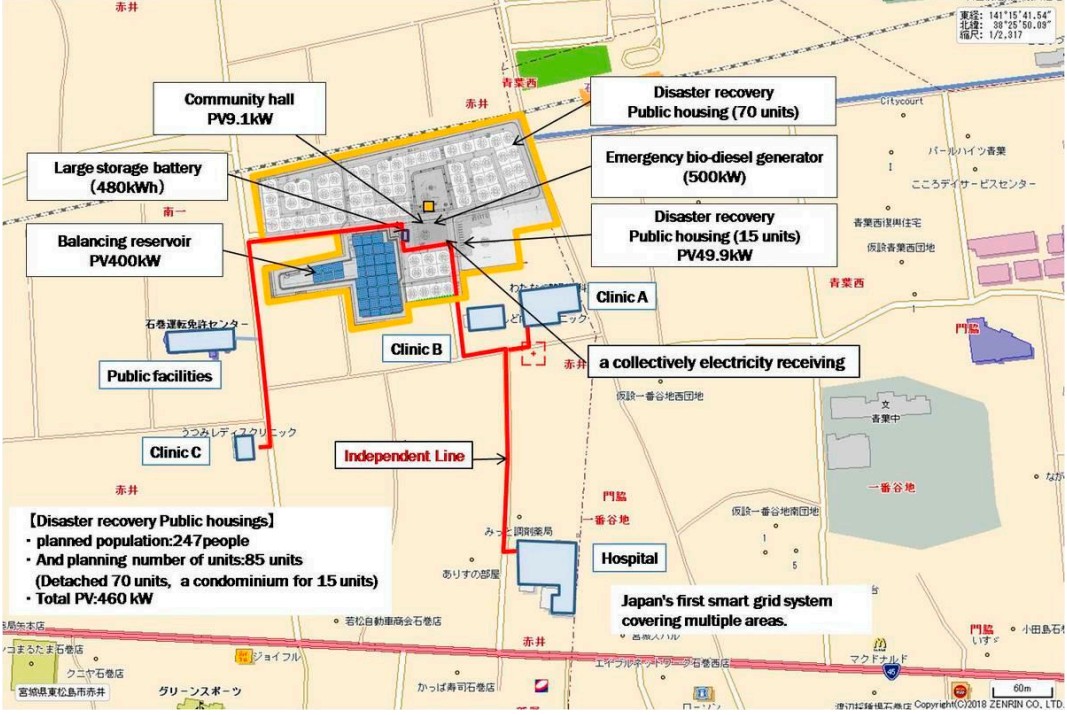

**Figure 3.** Higashi-Matsushima Smart Disaster Eco-Town [53].

### 3.1.3. Puerto Rico

In September 2017, Hurricane Maria hit Puerto Rico and caused approximately 1400 fatalities along with $90 billion in damage, including major destruction of critical infrastructures. and shortly after the next Hurricane Irma hit and cost another $50 billion, USD [55]. The island's power grid system was destroyed, and it took 11 months to recover power in all communities, which is recorded as the longest blackout in US history. This island-wide destruction of energy infrastructure triggered an ambitious post-disaster recovery plan to completely re-build Puerto Rico's electric power system. The "Build Back Better" plan was released in December 2017 and prepared by 14 agencies, including the US Department of Energy, the regional electric power authorities, research institutes, and private sectors [56]. With a budget of US$17.6 billion from government subsidies and private sector investments, the plan seeks to rebuild a more resilient power grid that is capable of surviving an upper Category 4 event (equivalent to Maria) by 2027. Specific technologies and strategies for increasing resiliency include building microgrids on critical infrastructures (Figure 4), modernizing transmission and distribution through smart grids, installing powerlines underground in high wind areas, increasing battery storage and onsite backup generation, combining heat and power systems, and implementing stricter vegetation management. The plan pays special attention to remote communities in order to help them quickly recover their power from a disaster through solar PVs, battery storage, and smart technologies. Not only intent on strengthening grid resilience, the plan also aims at a more aggressive energy transition to renewable energy by resetting their target to reach 50% (4000 MW) by 2035 and 100% by 2050, mainly through solar and wind. Prior to Hurricane Maria, Puerto Rico's renewable energy goal was to reach 20% by 2035. Throughout 2018, extensive stakeholder involvement encompassed the 41 key stakeholders across the island. The recommendations that emerged included: promoting self-sufficiency and credibility as its vision, an independent regulator with enforcement powers that promotes investor confidence and supports a sound prosumer sector, and a flexible regulatory framework and integrated resource plan that supports more innovation and renewable energy, and involving non-traditional players such as energy cooperatives and municipalities in the energy transition [57]. Puerto Rico Energy Commission (PREC) and Electric Power Authority (PREPA) have made further efforts in defining diverse users/ownership and system structures of microgrids including individuals and small cooperative entities [58,59]. The implementation of the Build Back Better plan and deployment of microgrids have been delayed due to the privatization process of PREPA's assets and leadership changes in the PREPA and the State Government. In October 2019, the state officially announced an upgraded 10-year "GridMod" plan with US$20 billion and stricter measures [60].

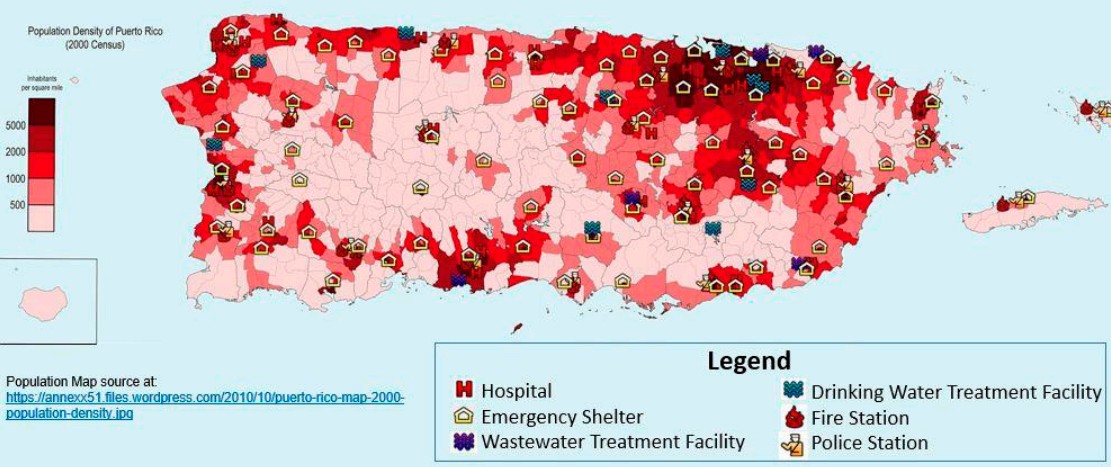

**Figure 4.** Hypothetical islanding of micro-grids for resiliency to support critical infrastructures in highly populated areas and remote communities in case of blackout [56].

### 3.1.4. Igiugig Village

This village has a year round population of 69 (as of January 2019) in southwestern Alaska, on the banks of the Kvichak River and Lake Iliamna. The population are largely Yup'ik Eskimos, Aleuts, and Athabascan Indians. The villagers live by subsistence fishing, hunting, and income from tourism associated with salmon runs in summer and hunting in the fall. The village is governed by the Igiugig Village Council. The Council has long sought to decrease village reliance on diesel powered electricity. In addition to residential and community services, electricity is vital for running the fish plant that provides much of the income for the community. The motivation for moving from conventional energy sources to renewables is two-fold. First, diesel fuel must be flown in across miles of tundra, forcing diesel prices to approximately $US7 per gallon, and fixing electricity prices at almost $US 0.80 per kilowatt hour (the national average is ~$US 0.10) [61]. Inclement weather and natural disasters can delay deliveries of fuel, placing the supply at risk. Second, the Village Council is responsible for maintaining the way of life for the native people and are committed to reducing the carbon footprint. Part of that commitment includes acting as a test site for emerging renewable technologies and energy systems. Starting in 2014, the Village Council has engaged with a commercial company to provide an alternative energy source that will supply the needs of the village, require relatively little expertise and resources to operate, and will protect the natural resources the village values. The company, Ocean Renewable Power Company (ORPC), is developing a hydrokinetic turbine that will reside under the surface of the river and generate power. The tidal device will feed into a shore-based microgrid which includes energy storage and the ability to bring existing diesel generators online if needed. The hydrokinetic device, RivGen, is a smaller version of an ORPC turbine tested in tidal waters in coastal Maine. Significant investments in the Igiugig RivGen project have been made by the US Department of Energy which has an interest in furthering the technology. However, additional critical funding has been supplied by the Alaska Energy Authority, with an Emerging Energy Technology Grant, and grants from private foundations, with the aim of helping the village become energy independent. The RivGen turbine was dedicated in a ceremony at Igiugig in July 2019 [62]. The village population joined the tidal developer as well as local and Alaska leaders to commit to the project (Figure 5).

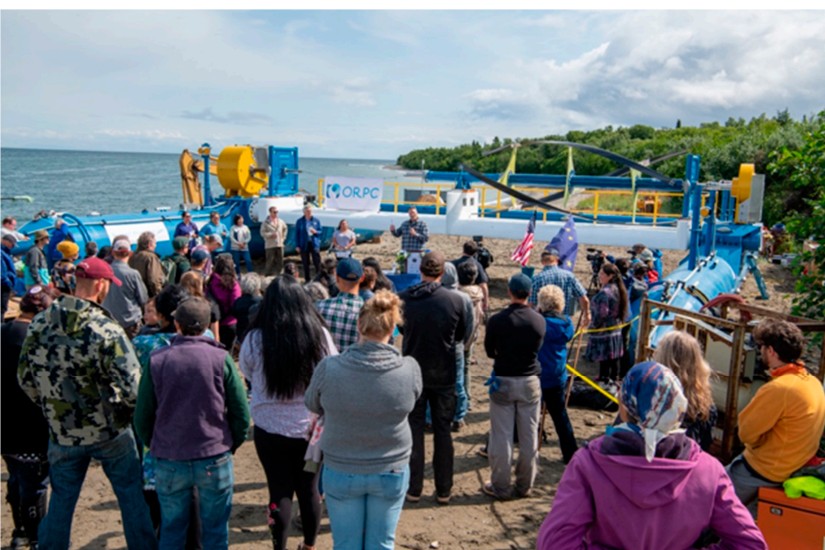

**Figure 5.** The Village of Igiugig, Ocean Renewable Power Company (ORPC), Department of Energy Alaska Governor Dunleavy, Senator Murkowski representative and others gather to celebrate the launch of the Igiugig Hydrokinetic Project and ORPC's RivGen® Power System on July 16 in Igiugig, Alaska. (Photo Credit: Alaska Governor's Office, Austin McDaniel).

### 3.1.5. Oregon Coastal Region

The portion of coastal Oregon that lies west of the Coastal Range is subject to strong Pacific storms and the area is prone to subsea earthquakes that may generate tsunamis. Most electricity in the Pacific Northwest region of the U.S. (including that in coastal Oregon) is regulated by the Bonneville Power Administration and is generated from hydropower, with some wind and fossil fuel sources. Portions of the Oregon coast are isolated from the neighboring portions of the national electricity grid (Figure 6). Following adverse conditions that may knock out this coastal grid, the necessary black start cannot be aided by adjacent grid segments, as is the standard procedure. These outages could leave the coastal population energy-vulnerable, potentially for days to weeks. Oregon state policy takes into account the use of renewable energy, including wind, solar, and marine renewable energy, for grid restart [63]. Coupled with concerns over coastal resiliency in the face of rising sea levels and climate change, and the need to plan for disaster recovery, plans for small scale, rapid deployment of wave energy devices have been developed [64,65]. The devices are planned to be easily deployed with locally available small vessels, without the need for heavy lift capabilities or specialized expertise. Adding this ready standby renewable capability to Oregon coastal areas will increase energy resilience as well as provide timely disaster recovery assets using low carbon sources in place of diesel generators supplied by state and federal disaster recovery efforts, to provide power after natural disasters. The most immediate needs such as freshwater desalinated from seawater and power for hospitals and emergency services, can be supplied with a series of small wave devices. These same devices, in conjunction with other renewables, can also supply the boost needed to restart the coastal grid. Actions to improve the coastal energy resilience in Oregon were brought together in 2019 under an Executive Order 17–20: Evaluation of Energy and Resiliency Efforts, that directs multiple state agencies to define and plan for statewide and coastal resiliency for all portions of the coastal grid, under conditions affected by natural disasters [66]. The Oregon planning effort has a strong focus on local stakeholder and community engagement, and ownership of new energy and coastal protection resources.

### 3.1.6. Qinghai Province

In China, poor air quality has been one of the top public health concerns in the past decade. The World Bank estimated in 2007 that 350,000 to 400,000 Chinese die prematurely each year due to air pollution [67]. Since the short-term success in mitigating air pollution during the 2008 Beijing Olympic Games, the public environmental awareness level has entered a stage of rapid rise in China [68]. Energy structure adjustment, to reduce coal and increase the renewable energy supply, has been the fundamental means to control air pollution [69]. In 2009, China announced that it will cut emissions as a percentage of economic output by 40 percent to 45 percent before 2020, compared with 2005, and increase the share of non-fossil energy in its primary energy consumption to around 15% by 2020. The environmental disasters like air pollution, trade frictions, and climate negotiation triggered the fast development of large scale renewable energy and the reduction of coal consumption. Specifically, in 2010, the working group of the Chinese Academy of Sciences pointed out the abundant and extensive solar energy resources in Qaidam Basin of Qinghai Province on the vast desert land; and nearly 5.9 million kilowatts of hydropower bases have been built in the upper reaches of the Yellow River, making Qinghai the ideal large-scale PV-hydropower national integrated energy base [70]. As shown in Figure 7, the 850 MW photovoltaic plant operating with hydropower in the complementary mode is situated 31 miles east of Longyangxia hydropower plant, creating the world's largest grid-connected PV power plant. The system generates electricity at a voltage of 330KV, which is transmitted to the grid through the Longyangxia hydropower station transmission lines [71–74]. The innovative design of a Water-PV complementarity energy generation system meets a number of goals including boosting renewable energy production, enhancing energy resilience and contributing to reduced air pollution levels.

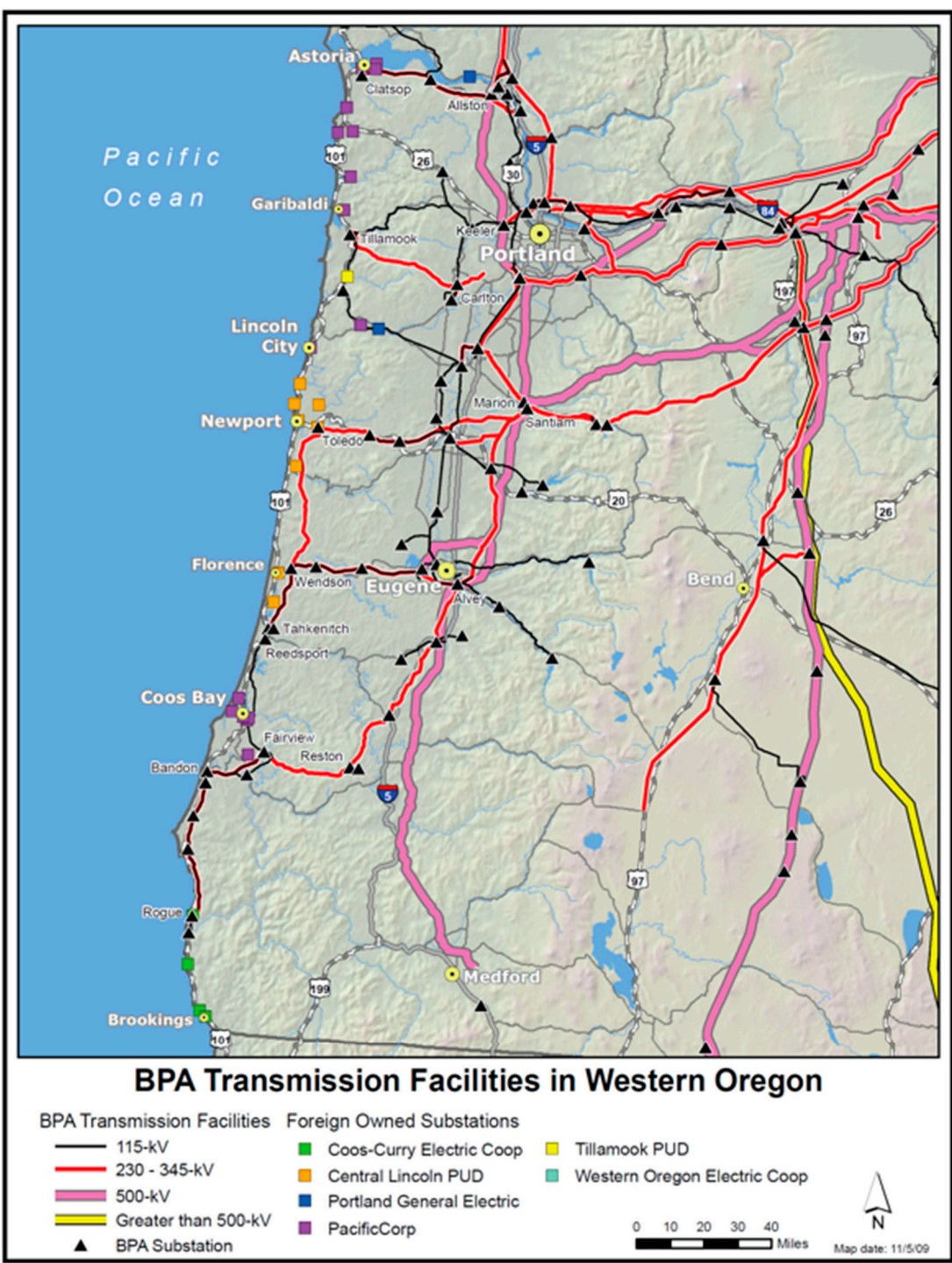

**Figure 6.** Bonneville Power Administration (BPA) grid connections in Oregon, with significant gaps in service on the Oregon coast. The southern gap near the California border is considered to be most at risk for extended power outages following coastal storms and tsunamis [63].

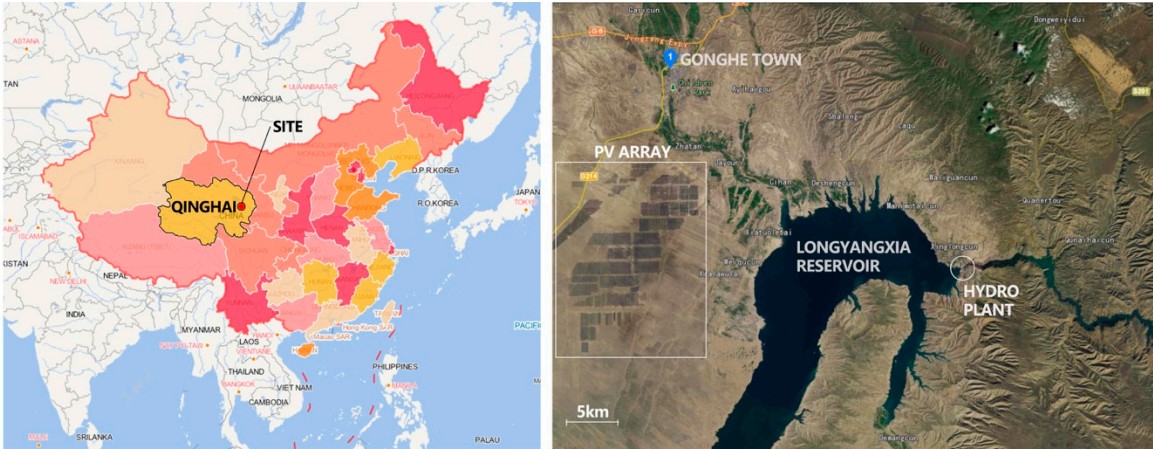

**Figure 7.** Location of the Longyangxia hydro-PV plant (Source: Prepared by Xin Wang).

### 3.1.7. Urawa-Misono District

In addressing the calls for carbon reduction through renewable energy while strengthening energy resilience to disasters and disruptions in the post-Fukushima era, a demonstration of a blockchain-based distributed energy system equipped with the digital grid router (DGR) and power interchange settlement was initiated in 2017 at Urawa-Misono district in Saitama City (population of district, 7476, as of February 2019) [75]. The digital grid can accommodate high penetrations of renewable energy and prevent cascading power outages through peer-to-peer trading of electricity [38]. This project is led by Digital Grid Corporation in collaboration with the University of Tokyo, Tateyama Kagaku group, Kansai Electric Co., Ltd., TEPCO Holdings, Hitachi IE system, NTT Data, Tessera Technology, and US-Design, with financial support provided by the Ministry of the Environment through the Low Carbon Technology Research, Development and Demonstration Program [75]. Peer-to-peer electricity exchanges are implemented among individual households including prosumers as well as consumers and a large shopping center [76]. Each prosumer has a DGR of 5 kW photovoltaics, a battery of 12 kWh, sub-grid selling and buying legs of 10 kW, and a digital grid controller (DGC) that communicates with buying and selling smart meters and DG net, whereas each prosumer has only a DGC [75]. As shown in Figure 8, a solar power system of 60 kW and DGRs have been installed at the shopping mall. Each DGC predicts demand and power generation and bids for selling and buying in units of 30 min for 24 h ahead, with settlements on the first-come, first-served basis. Balancing of power generation and consumption is automatically achieved in a decentralized manner based on a highly secure mechanism for a relatively small maintenance cost. This blockchain-based distributed energy system has relatively low adoption barriers, providing equal participation for any new entrants built upon the principle of distributed ownership.

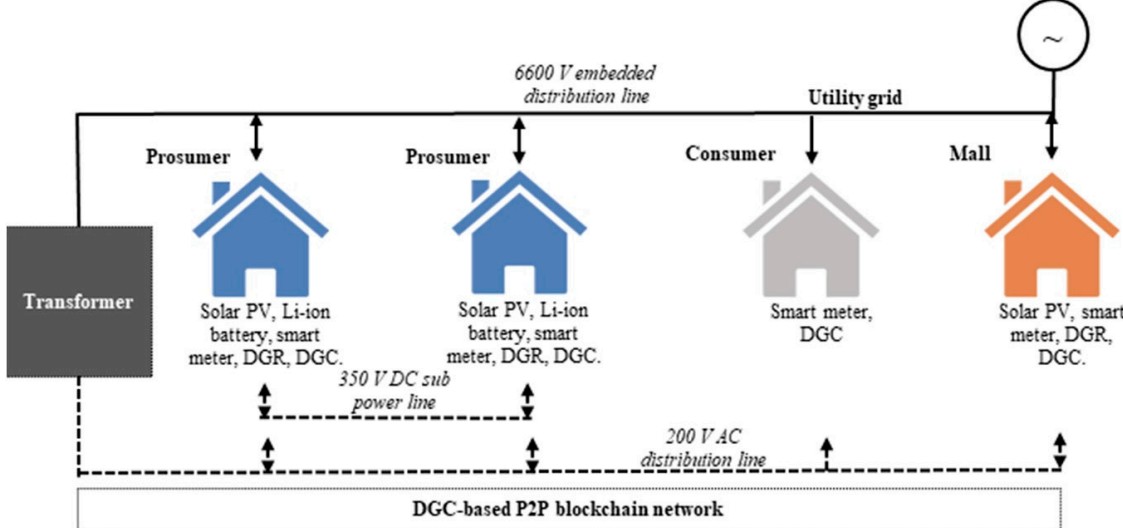

**Figure 8.** Blockchain-Based Distributed Energy System at Urawa-Misono District in Saitama, Japan [76].

### 3.2. Case Study Evaluation

Table 3 presents an evaluation of each case study in the context of their current status and the degree to which they have made progress in the transition to a resilience-oriented energy system. A total of 15 criteria are applied (sub-dividing further from the 11 criteria in Table 1) and the consistency of each case study with the definition of a resilience oriented energy system is evaluated. Although each case scores high, a simple count of the number of instances each case study meets the criteria reveals that the Urawa-Misono and Higashi-Matsushima cases are most consistent with the definition of resilience-oriented energy systems. The least consistent cases are the Igiugig Village (remote location lacking inter-operatability and interconnections with other networks) and Qinghai Province (large scale system structure lacking stakeholder engagement and user ownership). The subsequent discussion section reflects upon the degree to which natural disaster-triggered pathways emerged in each case study and whether preventative measures have been implemented in order to increase energy resilience in the face of future potential disasters.

**Table 3.** Evaluation of the case studies in relation to their consistency with resilience-oriented energy systems.

| Criteria | Characteristics of Resilient Energy Systems | South Australia | Higashi-Matsushima, Japan | Puerto-Rico, USA | Igiugig Village, USA | Oregon Coast, USA | Qinghai Province, China | Urawa-Misono, Japan |
|---|---|---|---|---|---|---|---|---|
| System structure | Cooperative-based networks of small-scale entities | ⊗ | ⊙ | ⊙ | ⊙ | ⊙ | ⊗ | ⊙ |
| | Bottom-up stakeholder engagement. | ⊗ | ⊙ | ⊗ | ⊙ | ⊙ | ⊗ | ⊙ |
| | Distributed power structure | ⊗ | ⊙ | ⊙ | ⊙ | ⊙ | ⊗ | ⊙ |
| Users/ownership | Owned by the community | ⊗ | ⊙ | ⊗ | ⊙ | ⊗ | ⊗ | ⊙ |
| Energy source/Environmental impact | Low-carbon, environmentally- friendly and clean renewable sources | ⊙ | ⊙ | ⊙ | ⊙ | ⊙ | ⊙ | ⊙ |
| Adaptability | Flexible energy systems adaptable to changes | ⊙ | ⊙ | ⊙ | ⊗ | ⊙ | ⊙ | ⊙ |
| Innovation | Transformation and innovation (e.g., ICT, virtual power plants, etc.) | ⊙ | ⊙ | ⊙ | ⊙ | ⊙ | ⊙ | ⊙ |
| Interconnectivity/interoperability | Feedback loops between different sub-systems | ⊙ | ⊗ | ⊙ | ⊙ | ⊗ | ⊙ | ⊙ |
| | Interconnectivity and interoperability | ⊙ | ⊙ | ⊙ | ⊗ | ⊙ | ⊙ | ⊙ |
| Modularity | An interconnected network of multi-scale systems | ⊙ | ⊙ | ⊙ | ⊗ | ⊙ | ⊙ | ⊙ |
| | Modular (autonomous), decentralized and distributed generation and transmission | ⊙ | ⊙ | ⊙ | ⊙ | ⊙ | ⊗ | ⊙ |
| Redundancy | Redundant facilities and redundant capacity (e.g., storage capacity) | ⊙ | ⊙ | ⊙ | ⊙ | ⊙ | ⊙ | ⊙ |
| Diversity | Diverse socio-economic and technological structures based on a variety of energy sources | ⊙ | ⊙ | ⊙ | ⊗ | ⊙ | ⊙ | ⊙ |
| Public support | Public and governmental support | ⊙ | ⊙ | ⊙ | ⊙ | ⊙ | ⊙ | ⊙ |
| Balance | Balancing efficiency with diversity and redundancy | ⊙ | ⊙ | ⊙ | ⊗ | ⊙ | ⊙ | ⊙ |
| Overall Score | Score from a max total of 15 | 11 | 14 | 13 | 10 | 13 | 10 | 15 |

Key: ⊙—Consistent, ⊗—Not consistent.

## 4. Discussion

This article has presented an analysis of seven case studies in four countries against a set of criteria that exemplify the characteristics of resilience-oriented energy systems, with particular focus on how natural disasters act as triggers to significantly destabilize conventional energy systems and lead to the emergence of new energy pathways. Each case study, to differing degrees, reveals that responding to, and preparing for disasters may allow subnational regions to begin a transition towards a more distributed, diverse, flexible, innovative energy system consisting of low carbon sources of generation. One significant difference among the case studies relates to the issue of scale. At the very local level, it is much more likely that the community will gain direct ownership and receive considerable direct benefits from a new, resilience-oriented energy system. However, the scope for innovation, efficiency, inter-connected networks, interoperability, and built-in redundancy may be greater at large scales of urban communities, regions and states. As yet, we acknowledge that few totally resilience-oriented energy systems exist, but it is clear from the case studies that a significant trigger event, like a natural disaster, can accelerate local communities and regions along the pathway towards energy resiliency and deep decarbonization. This is not to suggest, however, that local authorities simply wait for a disaster trigger before implementing measures to ensure the transition towards a resilience-oriented energy system. Rather, the earlier action is taken, the smoother the likely transition, with disaster situations potentially functioning as an accelerator of that transition under the right circumstances.

Moreover, transition analyses in the past have often neglected where transitions take place and the spatial configurations and dynamics of the networks within which transitions evolve [77]. The case studies presented here highlight the challenges associated with geographic location and scale of the energy system. This also raises questions about the role that local, regional, and state entities can play in shaping or directing socio-technical transitions, rather than being passive recipients of transition initiatives [78]. While the analytical framework applied in this research makes reference to the potential for the community in the resilience-oriented energy system to be more than passive consumers and to become owners, specifically at the small-scale community or district level, there is also the issue that for small-scale remote communities certain characteristics of resilience-oriented energy systems are difficult to manifest (e.g., interconnections of multi-scale energy systems).

The case studies highlight the complexity of the transition to low carbon, renewable and resilience-oriented energy systems, particularly in the context of the increased frequency of extreme weather and air pollution events including severe smog. They also shed light on the contested political environments in which transitions occur (for example in the South Australia case study) and the diverse players involved (global targets influencing national and local government policies, international technology providers, local electricity providers, regulators, financial institutions, cultural norms and consumers). One of the most interesting aspects relates to the manner in which emerging decentralized resilient (island-type) networks co-exist with the centralized grids supported by interconnections.

Energy resilience is commonly associated with physical and digital integrity of national electrical grids. However, significant numbers of people live in communities isolated from traditional grids, relying on a mix of locally generated power including generating electricity from diesel fuel, coal, or other petroleum products, combined or separate from renewable energy sources including solar, wind, wave, current, and biomass. For this reason, coastal, remote, and island communities may support microgrids to distribute power, but these systems commonly use less than 200 kW to 5 MW of power. In order to reach a sustainable level of energy resilience, these isolated coastal and island communities must address a range of issues including: security and safety of fuel supplies; installation, operation, and maintenance of conventional and renewable generation technologies with little outside expertise, specialized infrastructure, or capabilities; and must be prepared to be isolated for extended periods of time following natural disasters, conflicts, or seasonal extremes in weather. Many of these coastal and island communities are more susceptible to rising sea levels and other effects of climate change that will disproportionately affect their ability to secure and maintain energy production and use.

Conversely, the large-scale energy systems found in the Qinghai Province of China and South Australia cases, while capable of rapidly adopting renewable energy solutions, potentially suffer from the inability to provide more extensive energy user access to decisions on energy system deployment. The power to make energy-related decisions at this scale remains centralized, which may in turn undermine system resilience (a lesson learned the hard way in Japan after the Fukushima nuclear accident).

## 5. Limitations and Scope for Future Research

While this research has validated the notion of the resilience-oriented energy system and has illustrated how it can be applied to analyze specific case studies, there are a number of limitations with the conceptual framework presented in this article.

First of all, as identified in the research around transitions, there remains considerable difficulty in terms of distinguishing the triggering event (or subsequent political moves to initiate action), as well as how to identify when an energy system has reached a state of resilience or to what degree. In the case studies presented, the starting point is normally the date at which the natural disaster occurs (but not always). The endpoint is less clear, and there is no way to be sure that post-disaster pathways will remain and that some form of regressive trend would not occur (re-adoption of conventional energy systems). Indeed, a change of government in the South Australia case resulted in an "anti-renewable" energy political stance (even though the pathway proved unalterable).

Second, we accept that there is a tendency in this type of research to rely heavily on case study analysis of particular localities or technologies, while ignoring transitions with important political, cultural and societal aspects [79–81]. We also acknowledge a major limitation with our research in that it focuses only on cases in industrialized economies. A focus of technological challenges can be limiting, for example, in instances where there is no readily accessible renewable energy resource to take the place of conventional energy. This suggests that research focusing on locations vulnerable to disaster and associated rapid change in the "build back better" transitions may ignore wider debates about social systemic change, particularly the thorny questions of how to influence community behavioral patterns [82]. For instance, the analytical framework adopted in this article says little about key social science ontologies and about how power works in communities, a problem encountered by transition theories in general [83,84].

Third, there is the question of how post-disaster niche energy innovations can influence mainstream policies and practice [26], and on how to map the scope for alternative pathways for bottom-up promotion of low carbon adaptive energy resilience. This manifests as a concern about how to accelerate deep decarbonization and low carbon energy transitions, especially in circumstances of increasing vulnerability to extreme weather events and other natural disasters.

Within the context of low carbon, resilient urban transitions, many communities are struggling to unpack complexities around what low carbon or resilience actually means: who must be involved in the transition, how it should unfold and how to recognize the most effective pathways towards transition. A number of pathways have been described in the literature including (1) transformation; (2) dealignment/realignment; (3) technological substitution; (4) reconfiguration; and (5) system emergence [85]. Yet, we are still left trying to fathom what they mean on the ground in terms of implementation practice.

One of the main takeaways from this research for local policy makers should be that they implement, at the earliest opportunity, an audit of their current energy system against the criteria used to differentiate between conventional and resilience-oriented energy systems. This will help identify some of the current weaknesses with the design of the local energy system and provide insights on where changes are necessary to enhance resilience.

System emergence, for example, may be closest to the case studies examined in this research where the rapid transformation of a sub-national energy system is triggered by a disaster event. Clearly, the literature suggests emphasis should not be solely on technical and infrastructural shifts but should

also require dramatic reconfigurations in how we govern patterns of development with transformative political and economic implications [86]. Part of this challenge may relate to the need to integrate analyses of socio-technical transitions with other approaches such as quantitative systems modeling, socio-technical transition analysis, and initiative-based learning. This integration could facilitate an enhanced understanding of the governance challenges involved and determine how local, sub-national initiatives connect to national patterns and global scenarios [87]. Based on our work to date and that of other researchers, we recognize that future research possibilities involve questions of how innovation systems theories can make a contribution to the analysis of sustainability, low carbon and resilience transitions [88,89]. Finally, since this study draws on a limited number of case studies, future studies could include the possibility of increasing diversity of socioeconomic representation, as well as the number of cases, in order to improve the generalizability of the research results, especially with more case studies from developing economies.

**Author Contributions:** The original conceptualization of this research project was developed by Y.K., who coordinated the Urban Renewable Energy workshop at the 2018 APRU Sustainable Cities and Landscapes Conference, guided the research and facilitated the contributions from each individual author. The methodology for the research project was developed by all authors, and further enhanced with contributions from A.S. and B.F.D.B., Case studies were contributed by B.F.D.B., A.E.C., Y.K., M.Y. and X.W., Y.K. and B.F.D.B. contributed equally to this work and should be considered as joint first authors.

**Funding:** This research received no external funding.

**Acknowledgments:** We would like to acknowledge the feedback received from Jared Leader of the Smart Electric Power Alliance for his feedback on the Puerto Rico case study.

**Conflicts of Interest:** The authors declare no conflicts of interest.

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
