# Peer review of "Energy Transitions Towards Low Carbon Resilience: Evaluation of Disaster-Triggered Local and Regional Cases"

_sustainability, doi:10.3390/su11236801_

Round 1

Reviewer 1 Report

The paper analyzes seven case studies against a set of criteria that exemplifies resilient energy systems triggered by natural disasters. The study fits well with the journal's Special Issue "Indicators, Assessment Tools, and Rating Systems for Mainstreaming Sustainability in Urban Planning and Development" that aims to draw discussions on how development and application of indicators, assessment tools, and rating systems can support cities in their quest for sustainable development.

In general, the flow of ideas are well-presented and comprehensible for a wide-range of readers. In the introduction, the paper clearly describes the problem, literature gap, and higlights the academic contribution of the study. Individual cases studies are discussed briefly and analyzed. The paper further outlines the limitation of study and the directions for future research.

There are minor issues/comments that may add value to the paper.

1.  In section 3, the criteria for the selection of case studies are discussed which includes the types of renewable energy technologies, environmental disasters, geographic scales, and so on. On the other hand, it is mentioned from line 42 that the Asia and the Pacific are the most disaster-prone region experiencing extreme storms and cyclones, and severe flooding.

(a) The study used cases from four countries. Aside from the availability of data and the mentioned criteria, why didn't you include the countries that are mostly affected by natural disasters such as Myanmar, Sri Lanka, Dominica, Nepal, Peru, Vietnam, Madagascar, Sierra Leone, Bangladesh, Philippines, Thailand, etc.?

(b) In this study, the cases selected are from developed countries (special case for China) which have high capacity for R&D investment, climate-resilient infrastructures, and funding for low carbon technologies. Why didn't you include cases from developing countries which may have a very different disaster-related responses due to lack of resources?

2. The cases are evaluated in relation to their consistency with resilience-oriented energy systems. There are 11 criteria as described in tables 1 and 3.

(a) How each criterion is related to focus of the article which are the three dominant conceptualizations in terms of engineering, ecological, and socio-ecological resilience?

(b) The authors can make the criteria on both tables consistent. For example the level of establishment or public support; and the efficiency or balance.  

3. MINOR ISSUES

(a) In line 197, "a 50-year storm" could be "a once-in-50-years storm".

(b) In line 323, there is something missing on the sentence.

4. References

(a) Please follow the journal format for references. 

(b) Missing/wrong links for online resources (lines 535, 539, 548, 585, 595, 613, 680, 700, 724, 728, …)

(c) check the authors (lines 550, 565, 742,

(d) missing/wrong pages (lines 623, 721, 748)   

(e) missing title (line 715)  

(f) missing accessed date (lines 533, 644, …)

(g) DISHONEST access dates: 1 reference is same as your submission date (30 October 2019); six references are accessed in the FUTURE? (02 November 2019). 

Reviewer 2 Report

The paper deals with topical issues- low carbon resilience. However the paper needs improvement. Introduction needs to be improved. The aim of paper is not clear. The structure of paper needs to be addressed in the end of introduction. The input of paper is unclear. The methodology needs to be better presented. The assessment of case studies is also unclear. The criteria and scoring should be developed and assessed should be conducted based on these scores. The selection of case studies also needs more background. The discussion section needs improvement as well. The strengths and weaknesses or limitations of applied approach needs to be better addressed. Conclusions also need improvements by presenting findings of this paper in more concise way. The policy implications of conducted study need to be developed.

Round 2

Reviewer 2 Report

  Comments were answered.